# Optical Spectroscopic Detection of Mitochondrial Biomarkers (FMN and NADH) for Hypothermic Oxygenated Machine Perfusion: A Comparative Study in Different Perfusion Media

**DOI:** 10.3390/s25134031

**Published:** 2025-06-28

**Authors:** Lorenzo Agostino Cadinu, Keyue Sun, Chunbao Jiao, Rebecca Panconesi, Sangeeta Satish, Fatma Selin Yildirim, Omer Faruk Karakaya, Chase J. Wehrle, Geofia Shaina Crasta, Fernanda Walsh Fernandes, Nasim Eshraghi, Koki Takase, Hiroshi Horie, Pier Carlo Ricci, Davide Bagnoli, Mauricio Flores Carvalho, Andrea Schlegel, Massimo Barbaro

**Affiliations:** 1Department of Electric and Electronic Engineering, University of Cagliari, Via Marengo 2, 09123 Cagliari, Italy; 2Department of Inflammation & Immunity, Lerner Research Institute, Cleveland Clinic, Cleveland, OH 44195, USA; sunk2@ccf.org (K.S.); jiaoc@ccf.org (C.J.); satishs2@ccf.org (S.S.); yildirf@ccf.org (F.S.Y.); karakao@ccf.org (O.F.K.); crastag2@ccf.org (G.S.C.); eshragn@ccf.org (N.E.); takasek@ccf.org (K.T.); horieh@ccf.org (H.H.); schlega4@ccf.org (A.S.); 3Transplantation Center, Cleveland Clinic, Cleveland, OH 44195, USA; wehrlec@ccf.org; 4Department of Physics, University of Cagliari, Complesso Universitario di Monserrato, S.P. Monserrato-Sestu Km 0,700, 09042 Monserrato, Italy; carlo.ricci@dsf.unica.it; 5Medica Spa, Via Della Beverara, 46/d, 40131 Bologna, Italy; davide.bagnoli@medica-spa.com; 6Bridge to Life Ltd., Atlanta, GA 30096, USA; m.carvalho@b2ll.com

**Keywords:** mitochondrial molecular biomarkers, spectroscopy application, optic phenomena in perfusion solution, spectroscopic calibration, methodologies and protocols, optic sensor theoretical foundations

## Abstract

**Highlights:**

**What are the main findings?**

**What is the implication of the main finding?**

**Abstract:**

Ex situ machine perfusion has emerged as a pivotal technique for organ preservation and pre-transplant viability assessment, where the real-time monitoring of mitochondrial biomarkers—flavin mononucleotide (FMN) and nicotinamide adenine dinucleotide (NADH)—could significantly mitigate ischemia-reperfusion injury risks. This study develops a non-invasive optical method combining fluorescence and UV-visible spectrophotometry to quantify FMN and NADH in hypothermic oxygenated perfusion media. Calibration curves revealed linear responses for both biomarkers in absorption and fluorescence (FMN: λ_ex_ = 445 nm, λ_em_ = 530–540 nm; NADH: λ_ex_ = 340 nm, λ_em_ = 465 nm) at concentrations < 100 μg mL^−1^. However, NADH exhibited nonlinear fluorescence above 100 μg mL^−1^, requiring shifted excitation to 365 nm for reliable detection. Spectroscopic analysis further demonstrated how perfusion solution composition alters FMN/NADH fluorescence properties, with consistent reproducibility across media. The method’s robustness was validated through comparative studies in clinically relevant solutions, proposing a strategy for precise biomarker quantification without invasive sampling. These findings establish a foundation for real-time, optical biosensor development to enhance organ perfusion monitoring. By bridging spectroscopic principles with clinical needs, this work advances translational sensor technologies for transplant medicine, offering a template for future device integration.

## 1. Introduction

Organ transplantation is a critical treatment for patients with end-stage organ failure, and maintaining graft viability during the ischemic period is paramount for successful outcomes. In recent years, machine perfusion has emerged as an advanced method for organ preservation, offering the potential for the continuous monitoring of graft quality through the analysis of biomarkers in perfusion solutions [1]. While conventional parameters such as lactate, pH, and liver enzymes (ALT, AST, LDH) provide valuable but indirect information regarding metabolic status [2,3,4], there is a growing interest in using mitochondrial-derived biomarkers—specifically, flavin mononucleotide (FMN) and nicotinamide adenine dinucleotide (NADH)—as direct indicators of ischemia-reperfusion injury (IRI), shown in Figure 1, and mitochondrial dysfunction [5,6,7,8,9,10].

Prior studies have demonstrated the potential of the real-time fluorescence detection of FMN and NADH during organ perfusion [5,7,11]. However, a critical review of the literature reveals several limitations:**Incomplete calibration details:** Many works do not report the calibration curves or experimental parameters (e.g., integration times, instrument settings) necessary to reproduce and compare their results.**Limited detection methods:** Most studies rely solely on fluorescence measurements—even though only FMN typically exhibits a robust fluorescence response—while neglecting absorbance techniques, which may be more suitable for quantifying NADH, particularly at high concentrations where nonlinear effects and signal saturation occur.**Mismatch between calibration and measurement conditions:** In several instances, the calibration medium differs from the perfusion medium used in actual settings, which can compromise the reliability of the optical readout.**Inconsistent methodology across studies:** Varied excitation/emission wavelengths and different spectrometers are currently used, making cross-comparisons challenging. For instance, while some studies use 450 nm excitation for FMN detection, others deploy suboptimal wavelengths (e.g., 405 nm or 485 nm) without justifying their choice, leading to potentially longer integration times and added complexity in sensor design.

Table 1 provides a comparative overview between the main studies in literature and our work, highlighting differences in detection methods, experimental conditions, and validation parameters.

In light of these challenges, our study presents a comprehensive spectroscopic investigation of FMN and NADH in various perfusion media that are routinely used—or intended for future use—in organ perfusion. Specifically, we examine both fluorescence and absorbance modalities across a wide range of concentrations and in diverse media formulations. Our work systematically evaluates:The linearity and detection limits of calibration curves,The influence of excitation wavelength on signal strength and specificity,The impact of media composition on spectral profiles, andThe reproducibility and robustness of the optical measurements.

By addressing these issues, our research lays the theoretical foundation necessary for the development of real-time optical biosensors. In doing so, it provides key insights for sensor designers, such as the importance of performing media-matched calibration, choosing optimal excitation wavelengths to reduce integration time, and combining fluorescence with absorbance detection to overcome limitations in quantifying NADH. These contributions directly respond to the current gaps in the literature, as summarized in recent studies [5,7,9,10,14].

The remainder of the manuscript is organized as follows: Section 2 describes the experimental materials, methods, and instrumentation; Section 3 presents our results along with a detailed comparison with the existing literature; and Section 4 concludes with a discussion of the implications for future sensor development and potential clinical applications.

## 2. Materials and Methods

We have used a conventional bench spectrophotometer and a bench spectrofluorometer. Since perfusion can involve liver, kidneys, lungs, and heart, we focused on the perfusion solutions used in such cases, namely Belzer MPS^®^, Celsior^®^, Custodiol^®^ HTK, and IGL-1^®^. Experiments have been performed using riboflavin 5′-monophosphate sodium salt hydrate powder with purity ≥ 70% and NADH disodium salt grade II (purity at 98%, both from Sigma Aldrich, St. Louis, MO, USA). The two compounds were stored at −20 °C and 5 °C, respectively. Belzer MPS (Machine Perfusion Solution), Celsior, Custodiol, and IGL-1 solutions were prepared according to the recipes provided by the manufacturers and available in the literature [15] (Appendix A). Stock solutions containing FMN and NADH were also prepared according to usual weighing and dissolution procedures. All the chemicals involved in the preparation of the stock and perfusion solutions were purchased from Sigma Aldrich (Appendix A). The solution temperature was measured using a digital thermometer RS PRO RS 1720 (RS Components Ltd. Birchington Road, Corby, Northants, NN17 9RS, UK). Analytical samples were prepared utilizing a 50 mL volumetric flask (type A) and 100 μL of FMN collected with an ErgoOne^®^ Starlab micropipette. FMN concentration ranged between 0.12 μg mL^−1^ and 0.96 μg mL^−1^. Similarly, the NADH concentration was adjusted in the range between 20 μg mL^−1^ and 180 μg mL^−1^. The range of FMN and NADH concentrations under study was selected with the aim of developing a calibration curve suitable for quantifying these analytes in the perfusion fluid during hypothermic organ perfusion. Calibration curves for both absorption and fluorescence measurements were constructed by performing three independent replicates for each concentration point. For each analyte (FMN and NADH), absorbance–concentration and fluorescence–concentration relationships were recorded in triplicate. The resulting calibration curve corresponds to the average of the three individual linear fits obtained from these replicates. The choice of this range was based on the expected concentrations of FMN and NADH that might be released by the organ during the perfusion process [7,10,11,13,14,16]. Light sources were suitably shielded to avoid FMN and NADH degradation processes induced by irradiation during all the handling procedures. Samples were stored in a cool place, protected from light.

The absorption UV-Vis spectra were collected with a Jasco benchtop V-750 double beam UV-Visible spectrophotometer. The parameters configured in the spectrophotometer are provided in the Appendix A. Measurements were carried out in the spectral range of 200–800 nm using two Hellma^®^ high-performance quartz glasses, with a path length of 10 × 10 mm and a chamber volume of 3.5 mL. Fluorescence was studied using a Jasco FP-8550 spectrofluorometer (28600 Mary’s Ct, Easton, MD 21601, USA) equipped with a Hellma^®^ fluorescence cuvette made of high-performance quartz glass. Contamination was avoided by suitably cleaning the cuvettes and rinsing them with sample solutions. Glass or plastic Pasteur pipettes were used to inject samples into the cuvette. All the collected spectra have been compared with those collected from corresponding saline solutions of FMN and NADH with NaCl 0.9%. Full details of the measurement protocol are given in Appendix A. The collected data and figure were analyzed and processed using Python 3.8 in Spyder 4.2.

## 3. Results and Discussion

### 3.1. Absorption Spectra of FMN and NADH in Saline Solution in Different Conditions of Temperatures and pH

The absorbance and fluorescence spectrophotometry of FMN and NADH under different conditions has been extensively studied [17,18]. FMN absorbs in the near ultraviolet and blue visible light range between 320 nm and 500 nm due to its extended conjugated structure and the presence of alternating double bonds within the isoalloxazine ring. When the double bonds of the nitrogen atoms at positions 1 and 5 within the isoalloxazine ring system are irradiated with blue light, π bond electrons are excited to higher energy π* states. Notably, nitrogen atoms at positions 1 and 5 also serve as the redox centers of FMN. Upon the reduction of the molecule to the structures FMNH and FMNH_2_, the double bonds become saturated. Saturation results in the loss of absorption in the blue region and the consequent loss of the yellow color typical of flavin molecules. The absorption spectra of FMN and NADH with a concentration of 0.60 μg mL^−1^ and 180 μg mL^−1^, respectively, in saline solution at 5 °C, 20 °C, and 38 °C are shown in Figure 2.

The curves of FMN exhibit two maxima, respectively, at 373 nm and 445 nm. NADH has an absorbance behavior similar to FMN. The absorption spectra of NADH displays two maxima, in the UV region, at 260 nm and 340 nm. The peak at 260 nm is due to the adenine chromophore, whereas absorbance at 340 nm is related to the dihydronicotinamide moiety. In the temperature range explored (5–38 °C) no variation in the peaks’ positions has been observed for both coenzymes. It follows that FMN and NADH absorbance can be profitably detected under the hypothermic (5–10 °C), subnormothermic (20–25 °C), and normothermic (38 °C) conditions generally utilized for the machine perfusion of organs [19], using a calibration line referred for example at room temperature. Based on such evidence, we observe that temperature does not affect the spectrophotometric behavior of FMN and NADH. In Figure 3 we report calibration experiments at 20 °C.

For the FMN spectra, we determined the Limit of Detection (LOD) and the Limit of Quantification (LOQ). The LOD represents the minimum detectable signal, corresponding to the lowest concentration at which the signal can be reliably distinguished from noise. The LOQ, on the other hand, is the minimum concentration at which the analyte can be quantified with acceptable accuracy and precision. In our case, using our laboratory instrumentation, the LOD was determined to be 0.003 A.U., and the LOQ was 0.12 µg mL^−1^. At the LOQ concentration, the maximum absorption at the peak wavelength was approximately 0.003, a value close to the instrumental detection limit. Concentrations below 0.12 µg mL^−1^ could not be reliably detected with the current setup. In fact, below the signal corresponding to the concentration of 0.12 µg mL^−1^, it is not possible to distinguish between a true signal and the natural noise of the device. Generally, NADH concentrations in perfusate are higher than those in FMN [10,11,14]. Minimum concentrations of NADH are around 10 to 20 micrograms per milliliter, so there is no need LOD or LOQ for NADH, as it is already readily detectable in its naturally occurring concentration in the perfusate.

We also investigated the effect of pH on the absorbance spectra of FMN and NADH. Although the perfusate pH during hypothermic machine perfusion typically ranges between 6.5 and 7.5 [20], we extended our analysis to a broader pH range (Figure 4) to better characterize the spectroscopic behavior of these coenzymes under extreme conditions.

In saline solution, FMN remains spectrally stable in acidic environments (down to pH 2.5), showing no significant deviation from spectra acquired at physiological pH. Under strongly basic conditions (up to pH 12), however, FMN exhibits a characteristic blue shift, consistent with known pH-dependent spectral effects in flavins.

Conversely, NADH is highly stable in alkaline conditions (up to pH 12), while it undergoes degradation in acidic environments (pH 3–5), as evidenced by the disappearance of its distinctive absorbance peak at 340 nm.

Although such extreme pH values are not encountered during clinical machine perfusion, this extended analysis was performed to provide a comprehensive, phenomenological understanding of how pH affects the spectral properties of FMN and NADH. This knowledge may prove useful in evaluating sample stability or detecting abnormal conditions during perfusion protocols.

Furthermore, since we have seen that neither temperature nor pH change under physiological conditions affects absorption spectra, we can infer that temperature (5–38 °C) and the range of physiological pH change does not affect fluorescence spectra.

### 3.2. Absorption of Standard Perfusion Solutions

The quantitative evaluation of FMN and NADH concentrations during the machine perfusion of organs requires discrimination between the possible contributions to absorption and fluorescence coming from the analytes themselves and the perfusion solutions. For this reason, we measured the absorption spectra of the four standard perfusion solutions we selected. The curves are shown in Figure 5.

The four standard preservation solutions exhibit similar spectral behavior. The Belzer MPS, Custodiol, and IGL-1 solutions absorb strongly in the UV region, especially at wavelengths shorter than 300 nm, whereas Celsior medium absorbs substantially at wavelengths below 250 nm. All these aqueous solutions are primarily composed of sugars, organic and inorganic salts, amino acids, nitrogenous bases, starches, and buffering agents. The observed differences in absorption can be ascribed to compositional and concentration factors (Appendix A).

Figure 5a shows the absorbance of preservation solutions in the ultraviolet and blue spectral region approximately between 200 and 500 nm. Figure 5b presents a detailed comparative analysis of absorption spectra from various perfusion solutions, including the region where FMN absorbs. Figure 5b provides a magnified view of the absorption spectra of different perfusion media in the spectral region where FMN exhibits its characteristic absorption profile. The image compares the absorption spectra of various perfusion media with the absorption spectrum of a reference FMN solution in water at a concentration of 0.60 μg mL^−1^. The analysis reveals that the Belzer MPS solution exhibits notably higher absorbance compared to other perfusion solutions in the 340–500 nm spectral range. This comparative spectral analysis is crucial for understanding how the choice of perfusion medium can influence FMN measurements in clinical settings. In contrast, the absorbances of Celsior and Custodiol solutions are almost negligible compared with the Belzer MPS. IGL-1 shows a similar behavior to Custodiol.

When FMN, as well as NADH, was diluted in various perfusion solutions, we observed that the shapes of the absorption spectra were identical to those in saline solution. Aside from baseline corrections, the absorption values of FMN in saline solution were consistent with the light absorption values recorded in other perfusion solutions as depicted in Figure 6.

Figure 6 provides preliminary yet meaningful evidence supporting the chemical stability of FMN and NADH in the tested perfusion media. The absorption spectra exhibit consistent peak shapes and intensities across all solutions—Belzer MPS^®^, Celsior^®^, Custodiol^®^ HTK, and IGL-1^®^—when compared to the reference spectrum in water. The absence of spectral deviations, such as peak shifts, flattening, or intensity loss, suggests that no significant chemical degradation or interaction occurs between FMN or NADH and the constituents of the perfusion fluids.

To further support this observation, additional measurements were performed after allowing the FMN solutions to equilibrate in each perfusate for one hour at room temperature. The resulting spectra remained unchanged, indicating that any potential reactive phenomena are either absent or proceed at a rate negligible within the timescale relevant to machine perfusion protocols.

### 3.3. Fluorescence of FMN and NADH in Standard Perfusion Solutions

Fluorescence of FMN and NADH in saline solution is well documented [21,22]. FMN and NADH exhibit their fluorescence respectively in the regions between 500 nm and 600 nm and between 400 and 600 nm. The heat-map fluorescence spectrum of FMN in saline solution at a concentration of 0.36 μg mL^−1^ is shown in Figure 7a.

FMN exhibits significant fluorescence in the 520–560 nm range, with a maximum peak at approximately 538 nm. The emission resulting from an excitation wavelength of 455 nm is more intense compared to an excitation wavelength of 445 nm. Although this slight shift in excitation wavelength is normal in the presence of an absorption at 445 nm, it results in the two distinct calibration curves shown in Figure 7b. Data suggest that the sensitivity is highest when FMN fluorescence is excited at a wavelength of 455 nm. Consequently, we used this wavelength in subsequent experiments to evaluate the fluorescence of FMN in standard perfusion solutions. In this regard, it is worth noting that different excitation wavelengths are reported in the literature as well as that different emission wavelengths have been taken as reference for the quantitative or qualitative evaluation of FMN. The use of different excitation and emission wavelengths can raise significant difficulties in the comparison of clinical measurements obtained using different wavelengths [5,6,7,8,9,10,11,12,13,14,15]. For instance, the excitation light intensity, integral time, and other parameters being the same, the light emission at 538 nm due to excitation at 455 nm is almost twice as high as with excitation at 485 nm. It follows that induced emission from 485 nm can be obtained only if the light intensity or integral time are increased. In turn, this can give rise to uncertainties in measurement and control. For example, increasing the integral time can significantly burden data processing on normal devices. These practical aspects also need to be taken into serious consideration for in-line perfusion measurement.

The fluorescence of FMN in saline solution, as well as in Belzer MPS, Celsior, Custodiol, and IGL-1 perfusion solutions, along with the interference caused by these perfusion media, is illustrated in Figure 8.

The key observation is that the intensity of FMN fluorescence is affected by the composition of the perfusion solution and that the shape of the fluorescence spectra of FMN remains consistent across different perfusion media. This variation could be attributed to potential interactions between FMN and the compounds in the perfusion solutions, which may alter FMN’s electronic structure. This suggests that FMN does not undergo chemical reactions with other species present in the perfusion solutions as demonstrated by Figure 8. This conclusion is supported by the fact that the shapes of the spectra remain consistent across all perfusion media. As shown in Figure 5, different perfusion solutions exhibit varying capacities to absorb light in the 400–490 nm range. For instance, the Belzer solution absorbs light in the blue region, whereas water and Celsior do not, allowing all available photons to be absorbed by FMN. In contrast, when FMN is dissolved in perfusion solutions, such as Belzer, a portion of the photons is absorbed by the medium itself, reducing the number of photons available to excite FMN. This decrease in excitation photons results in reduced FMN emission. However, in other media like IGL-1 and Custodiol, the reduction in FMN fluorescence may not solely result from photon absorption but could also be influenced by solvent effects, as indicated by the emission spectra shown above. While the effects of solvents on flavins are well-documented in the literature [23], there is a notable absence of comprehensive studies specifically addressing FMN interactions in these types of media. Further investigation is warranted to elucidate the precise mechanisms underlying these observed phenomena and their implications for analytical applications. Consequently, the varying fluorescence of FMN in different perfusion media generates different calibration curves (Figure 9): the fluorescence emission of FMN in Belzer MPS solution is significantly lower than the one in Celsior or saline solutions. Interestingly, we observe that FMN in Custodiol solution exhibits significantly lower fluorescence compared to Belzer MPS solution, despite the negligible absorption of Custodiol in the 440–470 nm range. Furthermore, FMN in IGL-1 solution also shows emission similar to that of Belzer MPS, even though the absorption of IGL-1 in the blue region is slightly higher than that of Celsior and Custodiol.

Data in Figure 9 clearly show that the saline solution and Celsior solutions give rise to almost identical spectral behavior, with no interference with FMN. It follows that the saline solution of FMN can be used to calibrate the spectrophotometer if the Celsior solution is used as the perfusate. Conversely, such a solution is not viable in the case of Belzer-, IGL1-, or Custodiol-based perfusate. In general, similarities in the spectral behavior of calibration medium and perfusate, such as saline solution and Celsior, allow for reducing error bars in the concentration estimates. The full spectra of FMN in different perfusion media and error bars values are reported in Appendix A.

NADH has a fluorescence emission that ranges from 400 nm to 600 nm, and the experiments focused on the fluorescence induced at different excitation wavelengths. One representative fluorescence spectrum is shown in Figure 10.

Unlike FMN, the fluorescence spectrum of NADH exhibits greater complexity, as the shape of the spectrum is affected by both the excitation wavelength and the concentration. It is observed that as the concentration of NADH increases, the spectrum changes shape; in particular, for low concentrations, the excitation that generates the greatest fluorescence is at 340 nm, while for other concentrations the optimal wavelength to generate the maximum fluorescence is at 365 nm. This complicates the choice of the optimal excitation wavelength. A second peak in the deep ultraviolet region becomes evident at higher concentrations. Concentration affects not only the shape of the spectrum but also the calibration curve by generating a signal saturation effect as shown in Figure 11.

Although the curves are well defined, saturation effects appear at relatively high concentrations, around 100 μg mL^−1^, in every media. As a consequence, we observe non-linear calibration curves. This implies that, above 100 μg mL^−1^, the fluorescence signal is always the same and it is not possible to distinguish between different concentrations, making it impossible to accurately quantify NADH. Although concentrations normally found in perfusion media are below 70–80 μg mL^−1^, values above 100 micrograms could be recorded in some situations, especially in perfusate of livers. Detection by fluorescence would present an important limitation in that, in such situations, it would not be possible to be certain of the measured concentration.

As with FMN fluorescence, NADH fluorescence from Celsior and Custodiol media are comparable with the one from saline solution solutions. However, dissolved NADH can attain concentrations as high as approximately 100 times those of FMN. Therefore, although Celsior and Custodiol perfusion solutions absorb in the UV region, NADH can exhibit higher absorbance. In contrast, pure Belzer MPS has a higher absorbance than the Belzer solutions containing NADH, which reduces NADH fluorescence. Despite IGL-1 having similar behavior of Custodiol in absorption, NADH spectra in IGL-1 is the same as in Belzer MPS. As shown in Figure 12, fluorescence curves in different media almost perfectly overlap. Therefore, NADH does not interact chemically with the other components of the perfusion solutions.

Further details about NADH fluorescence spectra are given in Appendix A. Based on the results presented so far, the spectral behavior of perfusion solutions containing both FMN and NADH at different concentrations was investigated separately. Specifically, we measured fluorescence starting from an FMN concentration of 0.12 μg mL^−1^ and a NADH concentration of 20 μg mL^−1^. The two concentrations were increased simultaneously. A few representative three-dimensional fluorescence spectra for FMN + NADH saline solution solutions are shown in Figure 13. Fluorescence was exciting using a wavelength in the range between 300 nm and 390 nm in order to induce NADH emission in wavelength interval from 400 nm to 600 nm.

FMN and NADH share the UV absorption region between 320 nm and 390 nm. It can be seen from Figure 13a that, the FMN concentration being about 100 times lower than NADH, FMN poorly contributes to fluorescence. As the FMN concentration increases, a shoulder and then a peak become evident in the region between 500 nm and 530 nm due to excitation at wavelengths between 360 nm and 370 nm (Figure 13b). This can be explained by the excitation of FMN from NADH emission. Indeed, NADH emits light in the blue region where FMN absorbs. The increase in NADH concentration, and the associated increase in the NADH emission in the blue region, induces the excitation of FMN, eventually causing the growth of the peak in the yellow region. We have exclusively represented the behavior and interactions between FMN and NADH in physiological solution. The decision not to include interactions between FMN and NADH in other solutions (Belzer MPS, Celsior, Custodiol, IGL-1) was made because the interactions observable in water remain the same in other liquid media, with the only difference being lower intensities, as already highlighted in Figure 8 and Figure 12. Full details are given in Appendix A.

After discussing the mutual interference effects between FMN and NADH in fluorescence signals, we now focus on the absorption spectrum of a solution containing both coenzymes. As shown in Figure 2a,b, FMN and NADH share a common absorption range between approximately 300 and 400 nm, making it challenging to distinguish their individual contributions in this spectral region.

In Figure 14, we present the absorption spectrum of a mixture consisting of FMN at the highest concentration studied (0.96 µg mL^−1^) and NADH at the lowest (20 µg mL^−1^), with the goal of maximizing the potential interference of FMN on the NADH signal. This test aims to assess how much FMN might affect the quantitative estimation of NADH when absorption spectroscopy is used as the detection method.

The results show that FMN interference is minimal under this specific concentration combination. Applying the NADH calibration curve (Figure 3d) to the mixture’s spectrum yields an estimated NADH concentration of approximately 21 µg mL^−1^, slightly overestimated but still close to the true value. This demonstrates that, despite the spectral overlap, absorption spectroscopy can provide a reasonably accurate estimation of NADH, especially when NADH is present in significantly higher concentrations than FMN.

However, it is important to note that the accuracy of this estimation depends on the relative concentrations of the two coenzymes. In cases where FMN is more concentrated or NADH is present at lower levels, the uncertainty in the measurement may increase. From a clinical perspective, it is therefore essential to determine whether such uncertainty remains within acceptable limits for monitoring or diagnostic purposes.

Our approach thus offers a practical evaluation of the reliability of absorption spectroscopy for NADH quantification in the presence of FMN, contributing to a more informed assessment of its applicability in real-world clinical scenarios.

## 4. Contribution to the Development of Sensing Technologies

Our work also provides a crucial foundation for developing portable devices and automatic spectral recognition systems. The evolution of perfusion monitoring has been driven by the need for real-time, non-invasive sensing technologies. The Zurich Group’s portable perfusion-line-integrated device (2019, 2023) marked a breakthrough in FMN/NADH detection, yet fundamental challenges remain for clinical translation: uncharacterized medium-dependent fluorescence effects and unoptimized excitation parameters for portable systems. Three key limitations emerge from literature:Methodological inconsistencies: Studies like those by Müller et al. (2019) and Wang et al. (2020) [5,7] lack critical details (calibration curves, integration times) essential for replicating sensing protocols.Device optimization gaps: Huwyler et al.’s [11] device focused on a narrow NADH range, missing the saturation effect we observed >100 µg mL^−1^ (Figure 11)—a critical threshold for sensor dynamic range design.Validation shortcomings: Sousa Da Silva/van de Leemkolk [9,10] omitted instrumentation details, compromising their utility as references for sensor calibration.


**Direct Implications for Optical Sensor Development**


Our findings provide actionable insights into sensing systems:Medium-specific calibration (Figure 8, Figure 9, Figure 10, Figure 11 and Figure 12) proves that perfusion solution composition dramatically affects fluorescence, requiring on-device calibration profiles for clinical accuracy.Excitation wavelength flexibility (Figure 9, Figure 10 and Figure 11) suggests future devices should incorporate adjustable LEDs (340–450 nm) to accommodate varying analyte concentrations.FMN–NADH fluorescence interaction (Figure 13) demands embedded spectral deconvolution algorithms to prevent crosstalk in multi-analyte detection.Acknowledging that the detection by fluorescence of NADH has a limitation due to signal saturation effect (Figure 11).The implementation of machine learning algorithms capable of recognizing the coenzyme signal from other signals and consequently calculating the corresponding concentration.

Our work addresses these gaps while providing a roadmap for next-generation optical sensor development. Optical data is the mandatory first step in developing devices since no sensor can be designed without these preliminary results.

## 5. Conclusions

We extensively investigated the spectrophotometry of FMN and NADH in standard perfusion solutions to construct calibration curves for quantitative evaluation during machine perfusion. We propose a simple, accurate, and precise calibration method applicable to both benchtop and inline spectrophotometers. This approach is adaptable to various perfusion liquids, including Steen^®^ and Perfadex^®^, and enables cost estimation related to calibration procedures.

Our findings show that FMN and NADH maintain consistent absorption and fluorescence spectra regardless of the perfusion solution used. Neither pH nor temperature significantly affect their spectral behavior. However, interactions with certain perfusion fluids, such as Belzer MPS, IGL-1, and Custodiol, can reduce fluorescence—especially for FMN—without involving chemical reactions. In contrast, Celsior behaves similarly to saline solution and does not interfere with fluorescence intensity. These observations emphasize the need to perform calibration in the same perfusion medium used for clinical applications.

Based on the results obtained, the use of fluorescence spectroscopy with monochromatic excitation is recommended for the selective detection of FMN. Specifically, excitation at 455 nm proves to be highly specific for FMN, as NADH does not absorb at this wavelength. This allows for the acquisition of a clean fluorescence spectrum free from interference. In contrast, the absorption spectra of FMN and NADH may exhibit substantial overlap in the 300–410 nm range, depending on their relative concentrations. In our experimental results, this overlap was limited, and the estimation of NADH concentration remained within an acceptable range. However, in different scenarios—particularly when FMN concentrations are higher and NADH concentrations lower—the interference could increase, potentially affecting quantification accuracy. From a clinical perspective, it is often more important to monitor the overall trend of biomarker levels over time than to obtain an exact absolute value. Nonetheless, determining the clinically acceptable margin of uncertainty remains an important topic for future investigation and validation. However, the application of machine learning algorithms for classification and regression, properly trained on spectra from mixtures of FMN and NADH at varying concentrations, can effectively support the discrimination of their respective spectral contributions. This is particularly relevant given that FMN is typically present at lower concentrations than NADH.

Therefore, we suggest using fluorescence for FMN quantification and absorption for NADH quantification. If fluorescence is also to be used for NADH detection, we recommend employing optical filters that block spectral components beyond 480 nm, in order to isolate the fluorescence of NADH (which peaks at 465 nm) and prevent overlap with FMN fluorescence.

Finally, the adoption of a sequential excitation strategy represents an additional measure to minimize interference and ensure accurate and differentiated detection of the two biomarkers.

FMN’s fluorescence calibration curve remains linear and robust, even in the presence of moderate interferents, making it suitable for both hypothermic and normothermic perfusion. NADH, however, loses fluorescence linearity above 100 µg mL^−1^ and is only relevant during hypothermic perfusion. Interfering substances, such as blood or additives (e.g., antibiotics, albumin, vasodilators), must be considered, and future research should assess their effect on biomarker quantification.

Our method is instrument-specific but broadly transferable to other systems. This study lays the groundwork for innovative device design and real-time monitoring using machine and deep learning algorithms. Techniques such as k-NN, SVM, and neural networks can classify biomarker spectra, estimate concentrations, and detect the presence of interferents. For example, a model trained on FMN data in Belzer MPS could monitor perfusion in real-time, aiding clinical decisions.

From a spectrophotometric perspective, the optimal perfusion solution mimics saline and avoids interactions with FMN and NADH. We recommend designing new solutions that do not absorb in the 400–500 nm range and do not alter biomarker fluorescence. Among those tested, Belzer MPS and IGL-1 performed best, particularly for FMN and NADH fluorescence.

This article should be viewed not only as a study on FMN and NADH spectrophotometry but as a guide for developing or refining devices for machine perfusion. It answers key questions about the best detection techniques, excitation wavelengths, calibration protocols, and the integration of machine learning. For existing setups or new devices, this framework offers a reference for the future development of devices and standardization.

## Figures and Tables

**Figure 1 sensors-25-04031-f001:**
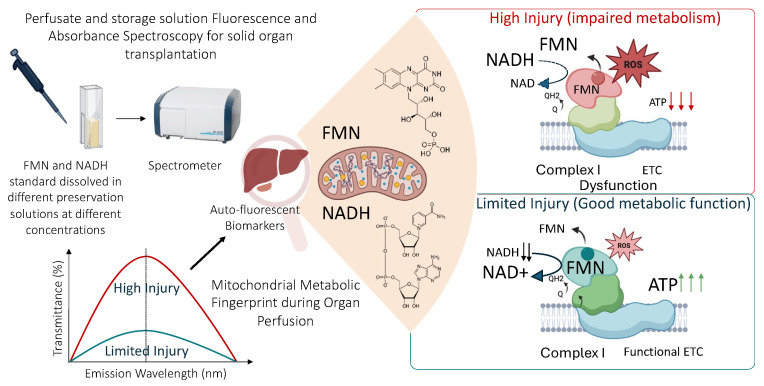
Overview of the underlying mechanism of FMN release and NADH accumulation and real-time spectroscopy for organ assessment.

**Figure 2 sensors-25-04031-f002:**
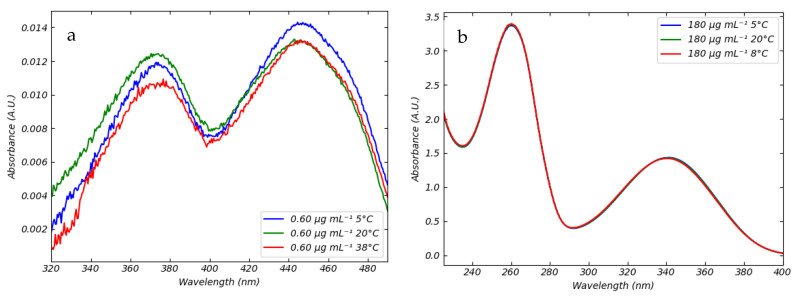
UV-Vis’s absorption spectra (**a**) FMN in saline solution at 5 °C (blue), 20 °C (green), and 38 °C (red). Data were obtained using a FMN concentration of 0.60 μg mL^−1^. (**b**) NADH at different temperatures at concentration of 180 μg mL^−1^.

**Figure 3 sensors-25-04031-f003:**
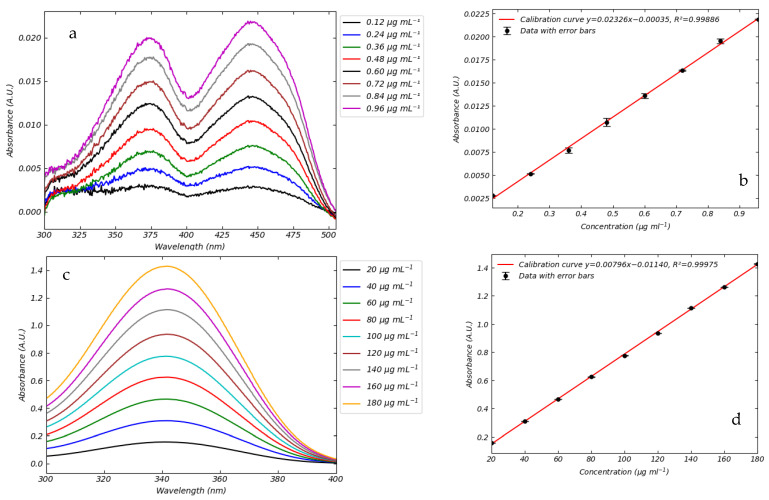
Absorption spectra of the coenzymes at different concentrations. Full absorption spectra of FMN and NADH in saline water in the 300 nm–400 nm range (**a**,**c**). Absorbance as a function of FMN and NADH concentration in saline solution. Data were collected using a wavelength of 445 nm and 340 nm for FMN and NADH, respectively (**b**,**d**). The best fitted line is also shown, and the corresponding error bar values are reported in Table 2.

**Figure 4 sensors-25-04031-f004:**
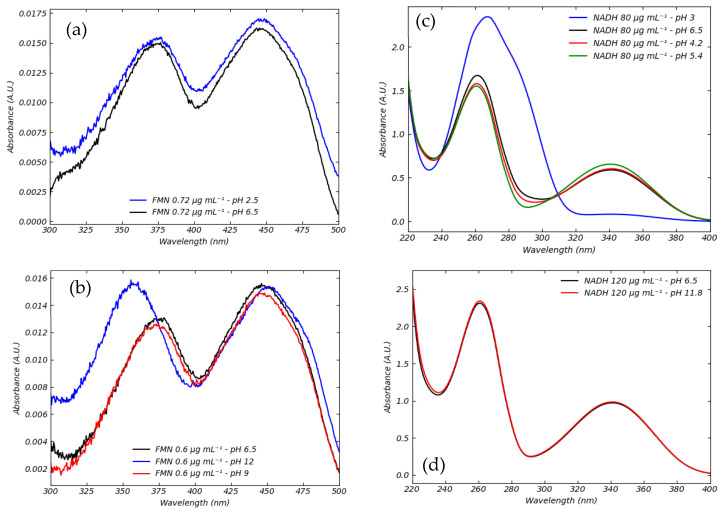
The effect of pH on FMN and NADH absorption spectrum. (**a**) The effect of pH acidic at 2.5 (blue) on spectra of FMN at 0.72 μg mL^−1^ compared to spectra at 6.5 (black). (**b**) The effect of pH basic at 9 (red) and at 12 (blue) on spectra of FMN at and 0.60 μg mL^−1^ compared to spectra at 6.5 (black). The effect of pH on NADH spectrum. (**c**) The effect of acidic pH at 3 (blue), 4.2 (red), and 5.4 (green) compared to spectrum at 6.5 (black) at concentration of 80 μg mL^−1^. (**d**) The effect of basic pH at 12 (red) compared to spectrum at 6.5 (black) at concentration of 120 μg mL^−1^.

**Figure 5 sensors-25-04031-f005:**
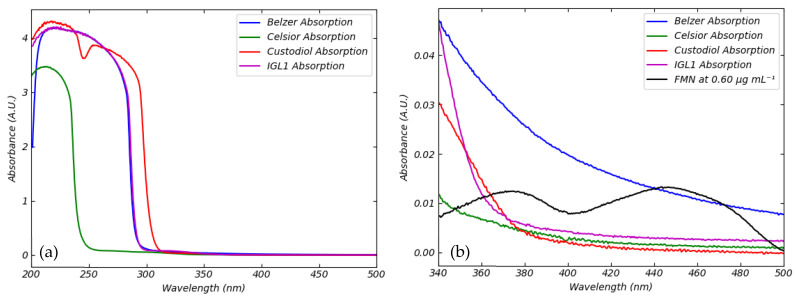
Absorption of perfusion media. (**a**) Absorption spectra of Belzer MPS (blue), Celsior (green), Custodiol (red), and IGL-1 (magenta) perfusion solutions. Plots show, in more detail, absorbance in the blue spectral region. (**b**) Absorption spectra of FMN at concentration of 0.60 μg mL^−1^ (black) compared to Belzer MPS (blue), Celsior (green), Custodiol (red), and IGL-1 (magenta) perfusion solutions.

**Figure 6 sensors-25-04031-f006:**
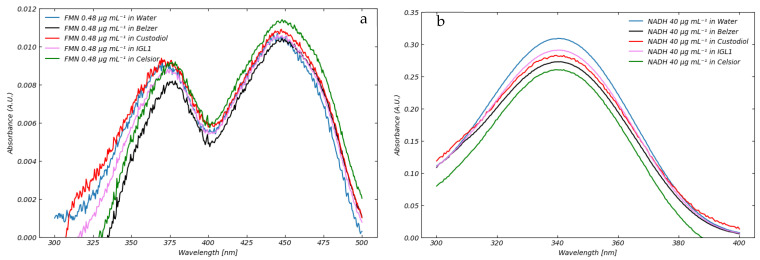
Absorption spectra of FMN (**a**) and NADH (**b**).

**Figure 7 sensors-25-04031-f007:**
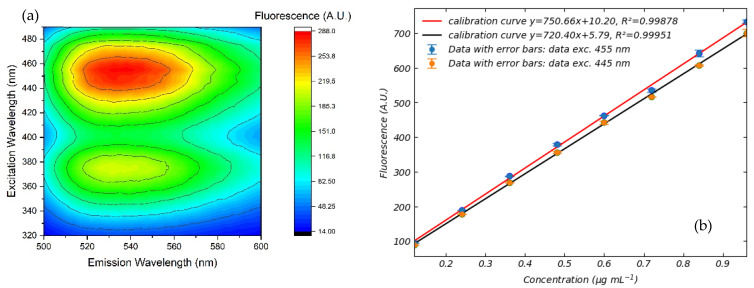
(**a**) Fluorescence spectra of FMN in saline solution at a concentration of 0.36 μg mL^−1^. (**b**) Calibration curves obtained using an excitation wavelength of 445 nm (black) and 455 nm for emission at 538 nm. Best fitting lines are also shown, and the corresponding error bar values are reported in Table 3.

**Figure 8 sensors-25-04031-f008:**
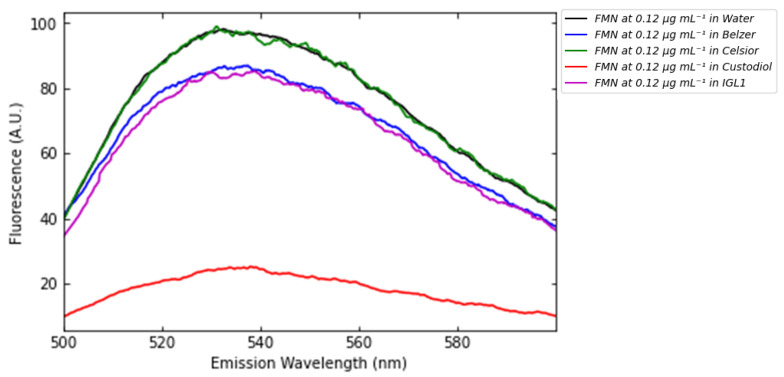
Fluorescence of FMN in saline solution (black), Belzer MPS (blue), Celsior (green), Custodiol (red), and IGL-1 (magenta) solutions at a concentration of 0.12 μg mL^−1^.

**Figure 9 sensors-25-04031-f009:**
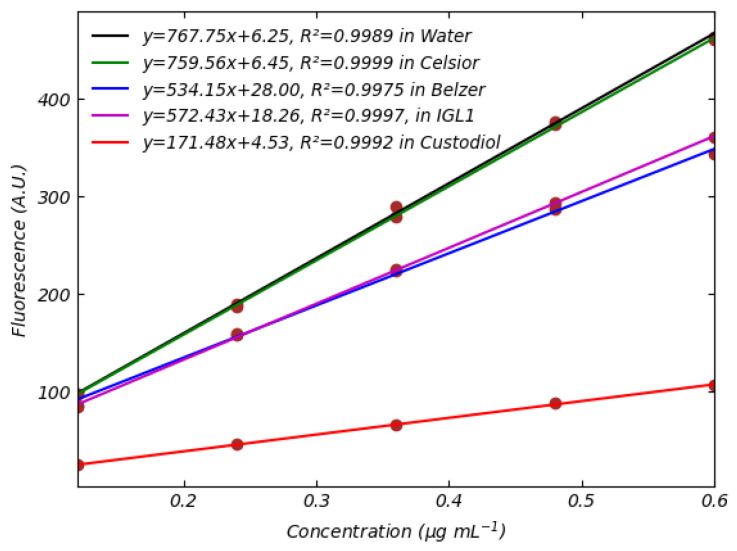
Fluorescence of FMN dissolved in saline solution (black), Belzer MPS (blue), Celsior (green), Custodiol (red), IGL-1 (magenta) solutions. Data were obtained using an excitation wavelength of 455 nm. Best-fitted lines are shown.

**Figure 10 sensors-25-04031-f010:**
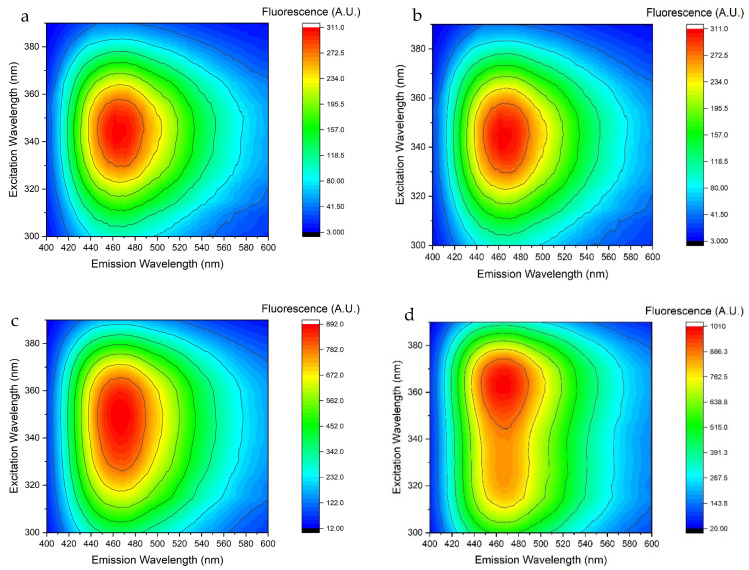
Fluorescence spectra of NADH in water at different concentration. (**a**) 20 μg mL^−1^, (**b**) 100 μg mL^−1^, (**c**) 140 μg mL^−1^, (**d**) 200 μg mL^−1^.

**Figure 11 sensors-25-04031-f011:**
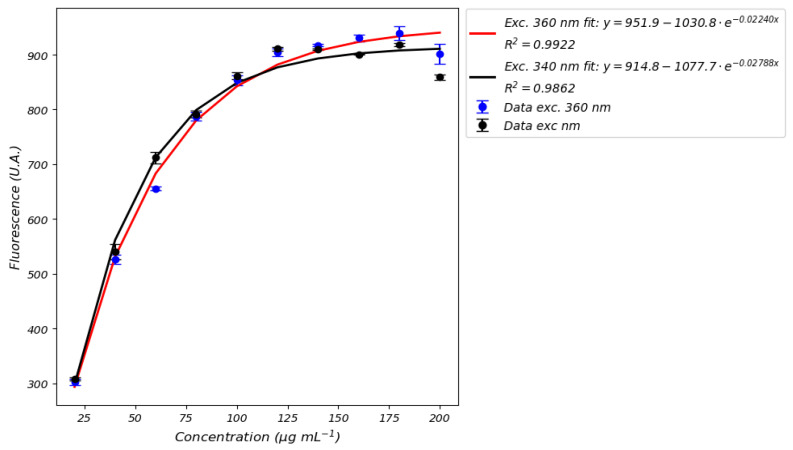
Calibration curves obtained using an excitation wavelength of 340 nm (black) and 360 nm (red). Emission values are referred to 465 nm. Error bars values are reported in Appendix A.

**Figure 12 sensors-25-04031-f012:**
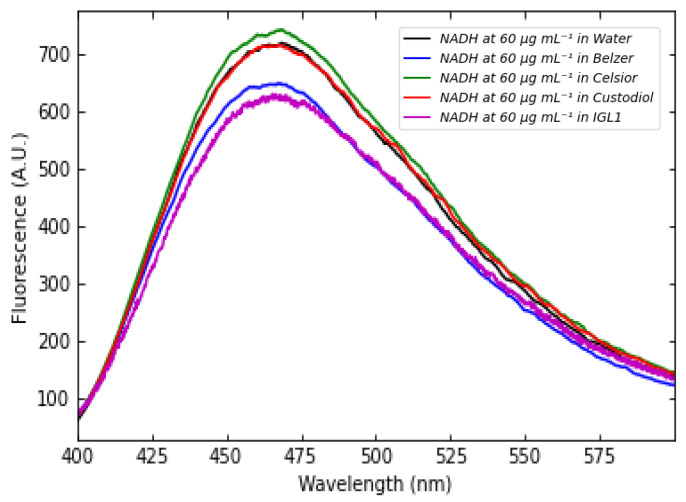
Fluorescence of NADH in saline solution (black), Belzer MPS (blue), Celsior (green), Custodiol (red), and IGL-1 (magenta) solutions at a concentration of 60 μg mL^−1^ excited at 345 nm.

**Figure 13 sensors-25-04031-f013:**
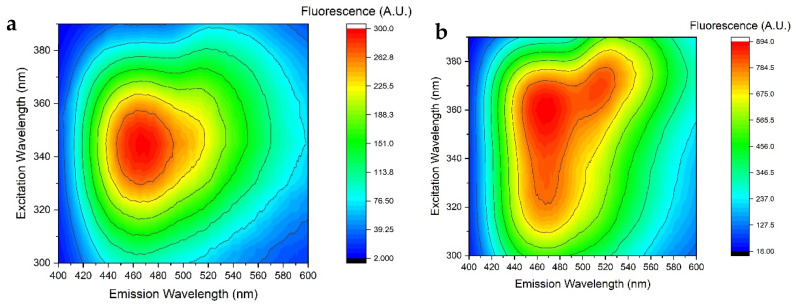
Fluorescence spectra of FMN + NADH saline solution solutions at FMN and NADH concentrations, respectively, equal to (**a**) 0.12 μg mL^−1^ and 20 μg mL^−1^ and (**b**) 1.08 μg mL^−1^ and 180 μg mL^−1^.

**Figure 14 sensors-25-04031-f014:**
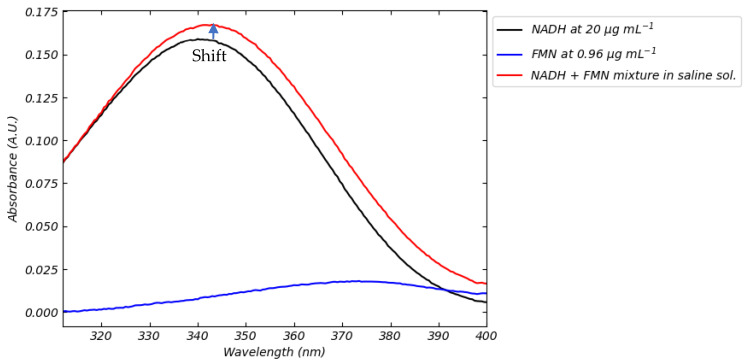
Absorption spectrum of the FMN+ NADH mixture and the effect of the potential interference of FMN.

**Table 1 sensors-25-04031-t001:** Comparative summary of clinical and preclinical studies reporting the detection and analysis of FMN and NADH, with an emphasis on contrasting findings with our study. The comparison includes: (a) the spectrometric detection technique; (b) the detection method utilized; (c) the influence of perfusion solutions on the measurements; (d) the consistency between the calibration medium and the organ perfusion medium; (e) the presence of a reported calibration curve; (f) the study typology; (g) Limits of Detection and Quantification (LOD/LOQ): represent the minimum concentration level that the instrument can accurately detect and quantify.

Ref.	Type of Analyte	Type of Method	Detection Method	Effects of Media Composition	Calibration Medium/Perfusion Medium	Calibration Curve	Reference Excitation/Emission Wavelengths	Spectrometer Used	Type of Study	LOQ and LOD
[5]	FMN	Fluorescence	Direct on perfusion line	Not reported	Not reported/Belzer	Not reported	Exc: 450 nm; Em: 500–600 nm	Portable	Clinical	Not reported in clinical context
[6]	FMN/NADH	Fluorescence	Direct on perfusion line/Perfusate Sampling	Not reported	Not reported/Belzer	Not reported	FMN-Exc: 450 nm; Em: 500–600 nm NADH-Exc: 360 nm Em: 465 nm	Portable	Preclinical	Not reported in pre-clinical context
[7]	FMN	Fluorescence	Perfusate Sampling	Not reported	Saline/Belzer	Not reported	Exc: 450 nm; Em: 525 nm	Microplate reader	Clinical	Not reported in clinical context
[9]	FMN	Fluorescence	Perfusate Sampling	Not reported	Not reported/Belzer	Yes	Exc: 485 nm; Em: 528 nm	Microplate reader	Preclinical	Not reported in preclinical context
[10]	FMN	Fluorescence	Perfusate Sampling	Not reported	Belzer/Belzer	Yes	Exc: 450 nm; Em: 500–600 nm	Microplate reader	Clinical	Reported for calibration: 0.00356 ng mL^−1^
[11]	FMN/NADH	Fluorescence	Direct on perfusion line	Not reported	Belzer/Belzer	Yes	FMN-Exc: 405 nm; Em: 530 nm NADH-Exc: 385 nm Em: 460 nm	Portable	Clinical	Reported for calibration: 2.01 ng mL^−1^
[12]	FMN/NADH	Fluorescence	Perfusate Sampling	Not reported	Not reported	Not Reported	Not Reported	Not reported	Preclinical	Not reported in pre-clinical context
[13]	Only FMN	Fluorescence	Perfusate Sampling	Not reported	Belzer/Belzer	Yes	Exc: 450 nm; Em: 525 nm	Microplate reader	Clinical	Reported for calibration: 7.9 ng mL^−1^
Our study	Yes	Fluorescence/Absorption	Preparation in laboratory of perfusion solution simulating perfusate	Yes	Fully described	Yes	Exc: 300–500 nm. Em: 400–600 nm	Benchtop	Methodologic	Reported for FMN calibration: 120 ng mL^−1^

**Table 2 sensors-25-04031-t002:** Standard errors for absorbance values used in FMN and NADH calibration curve (*n* = 3).

Concentration (μg∙mL^−1^)	Absorbance (Mean, A.U.)	Standard Error (A.U.)
0.12	0.002777	1.70 × 10^−5^
0.24	0.005138	2.63 × 10^−5^
0.36	0.007682	2.69 × 10^−4^
0.48	0.010725	4.20 × 10^−4^
0.60	0.013587	2.67 × 10^−4^
0.72	0.016343	7.40 × 10^−5^
0.84	0.019524	2.52 × 10^−4^
0.96	0.021873	6.51 × 10^−5^
20	0.155279	4.08 × 10^−4^
40	0.307527	6.66 × 10^−4^
60	0.465341	9.87 × 10^−4^
80	0.624495	1.39 × 10^−3^
100	0.775295	1.63 × 10^−3^
120	0.934276	8.05 × 10^−4^
140	1.114413	6.18 × 10^−4^
160	1.262550	4.58 × 10^−4^
180	1.426020	3.07 × 10^−3^

Standard errors (SE) of absorbance values were calculated and are reported. These SE values, although small (in the order of 10^−4^–10^−5^ A.U.), have been included to enhance statistical transparency and reproducibility.

**Table 3 sensors-25-04031-t003:** Fluorescence values and Standard Error (a) induced by 455 nm excitation, (b) induced by 445 nm excitation.

	Exc. 445 nm		Exc. 455 nm	
Concentration (µg∙mL^−1^)	Fluorescence (Mean A.U.)	Standard Error (A.U.)	Fluorescence (Mean A.U.)	Standard Error (A.U.)
0.12	94.4921	0.4302	88.7749	0.2351
0.24	189.2913	0.6099	177.7753	0.1832
0.36	287.3993	1.5043	268.9927	0.5976
0.48	379.3207	3.5497	356.0967	3.0412
0.60	462.7110	1.0385	441.9740	4.2942
0.72	535.1927	4.4261	516.2127	0.2915
0.84	642.2623	10.0139	607.7856	1.9467
0.96	733.7770	4.5954	700.7743	10.7981

## Data Availability

Essential data are contained within the article. Further data are available in the Appendix A. Data available on request.

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
