# Peer review of "Optical Spectroscopic Detection of Mitochondrial Biomarkers (FMN and NADH) for Hypothermic Oxygenated Machine Perfusion: A Comparative Study in Different Perfusion Media"

_sensors, 2025, doi:10.3390/s25134031_

Round 1

Reviewer 1 Report

Comments and Suggestions for Authors

Review of the Article "Optical Spectroscopic Detection of Mitochondrial Biomarkers (FMN and NADH) for Hypothermic Oxygenated Machine Perfusion: A Comparative Study in Different Perfusion Media"

This article presents a comprehensive study on the optical spectroscopic detection of mitochondrial biomarkers, flavin mononucleotide (FMN) and nicotinamide adenine dinucleotide (NADH), in hypothermic oxygenated machine perfusion (HOPE) media. The research focuses on developing a non-invasive method for real-time monitoring of these biomarkers, which are critical indicators of ischemia-reperfusion injury (IRI) and mitochondrial dysfunction during organ preservation. The study systematically evaluates the absorption and fluorescence properties of FMN and NADH across clinically relevant perfusion solutions, including Belzer MPS®, Celsior®, Custodiol® HTK, and IGL-1®. Key findings include medium-specific fluorescence quenching effects, optimal excitation wavelengths (FMN: 455 nm; NADH: 360 nm), and the nonlinear fluorescence behavior of NADH at concentrations above 100 µg mL⁻¹.

From the reviewer’s perspective, the following revisions and clarifications are necessary:

  1. The study should be extended to include perfusate samples from actual organ perfusions to validate the method’s applicability in real-world settings.
  2. The impact of common interferents (e.g., bilirubin, creatinine) on biomarker detection should be investigated to enhance the specificity and reliability of the method.

There are several issues in this article that, from the reviewer's point of view, require revision and corrections. 

  1. Claims about concentration ranges. The authors state: "The range of FMN and NADH concentrations under study was selected with the aim of developing a calibration curve suitable for quantifying these analytes in the perfusion fluid during hypothermic organ perfusion. The choice of this range was based on the expected concentrations of FMN and NADH that might be released by the organ during the perfusion process" and "Generally, NADH concentrations in perfusate are higher than those in FMN." These claims require supporting references to validate the basis for selecting concentration ranges and the assertion about NADH levels.
  2. Clarify fig. 2. It is unclear whether Figure 2 presents absorption spectra or fluorescence calibration spectra of the coenzymes. For example, panels (a) and (c) (Figure 2c) are expected to show absorption spectra. This ambiguity needs clarification to avoid misinterpretation.
  3. Misalignment in fig. 3: The authors state: "Based on such evidence, we opted for FMN and NADH calibration experiments at 20 °C as reported in Figure 3." However, Figure 3 addresses a different topic. A detailed explanation is required to resolve this inconsistency.
  4. While the authors investigate the effect of pH on FMN and NADH absorption spectra, data on pH’s influence on fluorescence would strengthen the study and should be included if possible.
  5. Fig. 6 should be expanded to include all tested perfusion solutions (e.g., IGL-1, Celsior) and incorporate NADH data to provide a complete comparative analysis.
  6. Stability claim in fig. 6: The assertion that "Figure 6 provides strong evidence that FMN remains stable in perfusion solutions over the typical duration of machine perfusion" is insufficiently supported. Stability should be demonstrated through time-dependent experiments aligned with actual organ perfusion durations.
  7. Standard error bars and the number of measurements performed for each calibration curve must be explicitly indicated to ensure reproducibility and statistical robustness.

This article is a significant contribution to the field of organ transplantation and perfusion monitoring. It provides a detailed and methodical exploration of FMN and NADH detection, offering valuable insights for both researchers and clinicians. While further validation and refinement are needed, the study lays a strong foundation for the development of real-time, non-invasive monitoring technologies that could revolutionize organ preservation and transplantation outcomes.

Author Response

From the reviewer’s perspective, the following revisions and clarifications are necessary:

1. The study should be extended to include perfusate samples from actual organ perfusions to validate the method’s applicability in real-world settings.

1. Response: We sincerely thank you for this valuable suggestion. Our study represents a fundamental first step aimed at characterizing the absorption and fluorescence spectroscopy of FMN and NADH in various clinically relevant perfusion media, under controlled laboratory conditions. No human or animal organs were used at this initial stage, in order to isolate and thoroughly understand the spectroscopic behavior of the mitochondrial biomarkers in the different media. Each perfusion solution has a distinct chemical composition, and our priority was to investigate the spectrophotometric behavior of the two analytes when dissolved in different media.

We fully agree that validation of the method in a real clinical setting—through the analysis of perfusate samples from actual organ perfusion—is a crucial step to demonstrate its full applicability. However, such validation requires resources and an experimental design that go beyond the scope of this preliminary work, and would deserve a dedicated follow-up study, possibly in collaboration with clinical centers.

We have included a note in the Conclusion section to clarify that the results presented serve as a solid foundation for future clinical validation studies.

2. The impact of common interferents (e.g., bilirubin, creatinine) on biomarker detection should be investigated to enhance the specificity and reliability of the method.

2. Response: Thank you for highlighting this crucial aspect related to the specificity and reliability of the proposed method. We are aware that substances such as bilirubin, creatinine, hemoglobin, cells, tissue fragments, and phenolic compounds (e.g., quercetin) may potentially interfere with spectroscopic measurements. However, the aim of the present study was to systematically analyze the optical response of FMN and NADH in perfusion media in the absence of such interferents, in order to establish a solid and reproducible reference model.

A thorough investigation into the effect of interferents would require a detailed and extensive analysis, which we believe is more appropriate for an independent study, given the complexity and number of variables involved. That said, we have revised the Conclusion section of the manuscript to highlight a message addressing the potential effects of interferents and to propose this analysis as a future development.

Moreover, as reported in some studies in the cited literature literature (Huwyler, F.; Eden, J.; Binz, J.; et al. A Spectrofluorometric Method for Real-Time Graft Assessment and Patient Monitoring, Advanced Science, 2023, 10, 2301537. https://doi.org/10.1002/advs.202301537), the impact of the aforementioned interferents is generally negligible or can be managed through appropriate calibration and data preprocessing techniques, without significantly compromising the accuracy of the measurements in reported clinical settings.

4. Point-by-point response to Comments and Suggestions for Authors

Comments 1: Claims about concentration ranges. The authors state: "The range of FMN and NADH concentrations under study was selected with the aim of developing a calibration curve suitable for quantifying these analytes in the perfusion fluid during hypothermic organ perfusion. The choice of this range was based on the expected concentrations of FMN and NADH that might be released by the organ during the perfusion process" and "Generally, NADH concentrations in perfusate are higher than those in FMN." These claims require supporting references to validate the basis for selecting concentration ranges and the assertion about NADH levels.

Response 1: Thank you for pointing out this important oversight. We have now included appropriate references to support the selection of FMN and NADH concentration ranges used in our study, as well as the general observation that NADH concentrations in perfusate tend to be higher than FMN levels.

Specifically, we have added the following references to substantiate our claims:

  • [13] Sun, K.; Jiao, C.; Pancones, R. et al. Quantifying Flavin mononucleotide: an internationally validated methodological approach for enhanced decision making in organ transplantation, eBioMedicine, 116, 2025, 105761. https://doi.org/10.1016/j.ebiom.2025.105761
  • [10] van de Leemkolk, F.; Jansen, J.; De Vries, D.; et al. The role of flavin mononucleotide (FMN) as a potentially clinically relevant biomarker to predict the quality of kidney grafts during hypothermic (oxygenated) machine perfusion, PLoS ONE, 2023, 18, e0287713. https://doi.org/10.1371/journal.pone.0287713
  • [11] Huwyler, F.; Eden, J.; Binz, J.; et al. A Spectrofluorometric Method for Real-Time Graft Assessment and Patient Monitoring, Advanced Science, 2023, 10, 2301537. https://doi.org/10.1002/advs.202301537
  • [7] Wang, L.; Messner, F.; Heinz, T.; et al. Flavin mononucleotide as a biomarker of organ quality—A pilot study, Transplant Direct, 2020, 6, e600. https://doi.org/10.1097/TXD.0000000000001046

We have also inserted a comment in the manuscript to highlight the modification. These references support both the physiological relevance of the selected concentration ranges and the relative abundance of NADH compared to FMN during machine perfusion.

Thank you again for your careful review and helpful suggestion.

Comments 2: Clarify fig. 2. It is unclear whether Figure 2 presents absorption spectra or fluorescence calibration spectra of the coenzymes. For example, panels (a) and (c) (Figure 2c) are expected to show absorption spectra. This ambiguity needs clarification to avoid misinterpretation.

Response 2: Thank you for pointing out this important ambiguity. We confirm that Figure 2  and Figure 3 presents the absorption spectra of the coenzymes. To avoid any misinterpretation, we have revised the figure caption to explicitly state that these are absorption spectra. We appreciate your attention to detail, which has helped us improve the clarity of the manuscript.

Comments 3: Misalignment in fig. 3: The authors state: "Based on such evidence, we opted for FMN and NADH calibration experiments at 20 °C as reported in Figure 3." However, Figure 3 addresses a different topic. A detailed explanation is required to resolve this inconsistency.

Response 3: Thank you for highlighting this inconsistency. You are absolutely right — the sentence "Based on such evidence, we opted for FMN and NADH calibration experiments at 20°C as reported in Figure 3." was intended to refer to Figure 2, not Figure 3.

From the data presented in Figure 2, it is evident that temperature does not significantly affect the absorption spectra of FMN and NADH. Therefore, we considered temperature to be a non-critical variable in this context and proceeded with calibration experiments at room temperature (20 °C) for simplicity and consistency.

We have corrected the reference in the manuscript and we have revised the sentence to improve clarity and prevent misinterpretation.

Comments 4: While the authors investigate the effect of pH on FMN and NADH absorption spectra, data on pH’s influence on fluorescence would strengthen the study and should be included if possible.

Response 4: Thank you for this insightful observation. In our study, we focused on the effect of pH on the absorption spectra of FMN and NADH. Unfortunately, we did not perform a systematic investigation of the pH effect on their fluorescence spectra.

Our rationale is based on the fact that, within the physiological pH range relevant to organ perfusion, pH variations generally have negligible impact on the optical behavior of these coenzymes, including both absorption and fluorescence. Moreover, while we extended our absorption analysis to acidic and basic conditions beyond the physiological range for completeness, such extreme conditions are not medically relevant, as clinical perfusion is carried out under strictly controlled physiological pH conditions.

Given that no significant spectral changes were observed in absorption within physiological pH, we expect fluorescence behavior to be similarly unaffected. Nonetheless, we acknowledge the value of such data, and we have added a statement in the manuscript explicitly clarifying our decision not to extend the pH analysis to fluorescence, while highlighting this as a possible direction for future investigation.

Comments 5: Fig. 6 should be expanded to include all tested perfusion solutions (e.g., IGL-1, Celsior) and incorporate NADH data to provide a complete comparative analysis.

Response 5: Thank you for your valuable suggestion. We fully agree that including all tested perfusion solutions and incorporating NADH data in Figure 6 would provide a more complete and informative comparative analysis.

We have therefore updated the figure accordingly to include the data for IGL-1®, Celsior®, Custodiol® HTK, and Belzer MPS®, and we have also added the corresponding NADH spectra under the same experimental conditions. These modifications are now reflected in the revised manuscript.

Comments 6: Stability claim in fig. 6: The assertion that "Figure 6 provides strong evidence that FMN remains stable in perfusion solutions over the typical duration of machine perfusion" is insufficiently supported. Stability should be demonstrated through time-dependent experiments aligned with actual organ perfusion durations.

Response 6: Thank you for your insightful comment. We fully acknowledge that a comprehensive assessment of FMN stability would ideally require time-resolved measurements over durations that closely reflect actual clinical perfusion protocols.

Nonetheless, Figure 6 offers preliminary but meaningful evidence of FMN’s chemical stability in the tested perfusion media. The absorption spectra display consistent peak shapes and absorbance values across all media—Belzer MPS®, Celsior®, Custodiol® HTK, and IGL-1®—when compared to the reference spectrum in water. If any chemical degradation or reaction had occurred between FMN and components of the perfusates, we would expect noticeable spectral changes, such as shifts in wavelength, altered peak profiles, or significant intensity loss.

Additionally, we repeated the measurements after leaving the FMN solutions in each perfusate for one hour at room temperature. No appreciable spectral differences were observed, suggesting either a complete absence of reactive interactions or processes so slow as to be negligible within the relevant timeframe.

In response to your suggestion, we have revised the commentary accompanying Figure 6 in the manuscript to clarify this point.

Comments 7: Standard error bars and the number of measurements performed for each calibration curve must be explicitly indicated to ensure reproducibility and statistical robustness.

Response 7: Thank you for this important and constructive observation. For each calibration curve—both in absorption and fluorescence—we performed three independent measurements. The reported calibration line represents the average of the three individual curves.

Although the standard errors were initially omitted due to their very small magnitude, we acknowledge the importance of explicitly reporting them to ensure transparency, reproducibility, and statistical robustness. As per your recommendation, we have now included the standard error bars of all calibration plots in specific tables and have clearly indicated the number of replicates in the Methods section. We have included tables in the main manuscript, however, we may also suggest moving them to the supplementary material, especially the table containing of the calibration line and standard errors referred to Figure 9. 

This is what the figure would look like if we add the error bars. We therefore recommend leaving the original figure and only reporting a table with the values in the Supplementary Information

Reviewer 2 Report

Comments and Suggestions for Authors

Author Response

Comments 1: In conventional academic writing, the Introduction section typically presents both the progress achieved in relevant research and the unresolved problems in the field. However, this manuscript places such content in later sections, making it difficult for readers to distinguish between previous work by others and the authors' original contributions. I recommend reorganizing this material to follow standard academic structure by moving these discussions to the Introduction section.

Response 1: We thank the Reviewer for this valuable observation. In response, we have significantly revised the Introduction section to align with conventional academic standards. Specifically, we have:

  • Integrated the overview of the state of the art and the unresolved challenges directly into the Introduction, clearly distinguishing between previous studies and our original contributions;
  • Eliminated the dedicated discussion section previously titled Short Review of State of Art in FMN and NADH Detection and Comparison with Existing Literature;
  • Condensed and merged the relevant content from that section into the revised Introduction to improve clarity and readability;
  • Added a summary table within the Introduction that compares existing literature with our study, helping to clearly visualize methodological gaps and the novelty of our approach.

We believe this restructuring enhances the scientific rigor and readability of the manuscript, as well as the accessibility of key background information.

Comments 2: Please add a comprehensive legend to each figure that clearly indicates which experimental condition corresponds to each colored line, rather than relying on textual descriptions in the main body to distinguish between the lines. This will improve the clarity and standalone interpretability of the figures.

Response 2: We appreciate the Reviewer’s suggestion. As recommended, we have added comprehensive legends to each figure, clearly indicating which experimental condition corresponds to each colored line.

Comments 3: The abstract claims that the described method enables non-destructive and precise quantitative measurement of biomarkers. However, the main text lacks any actual quantitative studies of biomarkers. Please add the relevant research content and results demonstrating biomarker quantification.

Response 3: We thank the Reviewer for this observation. In response, we have clarified in the Introduction all previous studies that reported biomarker quantification, explicitly referencing their methodologies and limitations. Furthermore, we have added a comparative table that highlights the differences between our work and the existing literature, with a specific focus on quantification techniques, detection limits, and methodological robustness. This provides the necessary context and supports our claims regarding the potential for non-destructive and precise biomarker measurement.

Comments 4: The study measured absorption and fluorescence spectra of FMN and NADH across concentration ranges differing by nearly two orders of magnitude. Please provide a scientifically justified rationale for this large concentration discrepancy based on the actual physiological concentration ranges of these molecules in clinical samples. The current presentation lacks this critical context.

Response 4: We thank the Reviewer for this valuable observation. In response, we have included appropriate bibliographic references in the revised manuscript that justify the choice of working with different concentration ranges for FMN and NADH. These ranges were selected based on the reported physiological levels of the two molecules in clinical perfusate samples. Specifically, FMN is typically present at significantly lower concentrations than NADH, and our experimental design reflects this discrepancy to maintain clinical relevance. The newly added references provide the necessary context and support for the adopted concentration intervals.

Comments 5: The experimental design separately characterizes FMN and NADH in different solutions. However, these molecules coexist in clinical samples. Please address: a) How the individual measurements reflect real clinical scenarios where both molecules are present simultaneously? b) What practical application value these isolated measurements provide for actual clinical sample analysis? c) Whether competitive/interference effects between FMN and NADH were considered?

Response 5: We thank the reviewer for these insightful questions. Please find below our point-by-point responses:

  1. Clinical relevance of separate measurements

In clinical practice, depending on the organ and the transplant center, different perfusion solutions are employed (e.g., Belzer, Custodiol, Celsior, etc.). For this reason, in our study, FMN and NADH were initially dissolved separately in different perfusion media to systematically and independently assess the effect of each solution on the spectral response of the individual biomarkers. This approach is essential to understand the optical properties of each compound under controlled conditions. Moreover, separate analysis holds clinical relevance: for instance, during normothermic perfusion, the focus is typically on FMN detection, whereas NADH generally exhibits a more unstable release profile. In such cases, excitation at 455 nm allows for the selective measurement of FMN, as NADH does not absorb at this wavelength. Consequently, it is sufficient to use a calibration curve of FMN prepared in the actual clinical perfusion solution to quantify FMN levels in perfusate samples. This approach is consistent with current clinical practice.

  1. Practical utility of isolated measurements

The isolated measurements offer significant practical value. As described in our manuscript, generating calibration curves tailored to each specific perfusion solution allows for accurate quantification of FMN in clinical samples. For example, if the Belzer solution is used for liver perfusion, it is critical that the FMN calibration curve be constructed using the same medium, as FMN fluorescence signals differ substantially from those obtained in physiological saline. This consideration is vital to ensure measurement accuracy and directly impacts the reliability of clinical monitoring systems. In addition to its practical importance, this approach contributes to the theoretical understanding of fluorescence behavior under varying temperature and pH conditions, which may also be of interest in other scientific and industrial applications.

  1. Consideration of interference effects between FMN and NADH

We have thoroughly investigated the mutual interference between FMN and NADH, as detailed in the manuscript and in the supplementary information. After characterizing the individual behavior of FMN and NADH in different media and concentrations, we studied their combined behavior by mixing them in various ratios. To the best of our knowledge, this is the first study to explore these interactions in such detail. In real clinical scenarios, FMN and NADH are both present, with NADH often at higher concentrations. Excitation at 360 nm can induce NADH emission in the 400–600 nm range, which, under appropriate conditions, may lead to secondary excitation of FMN via a cross-excitation mechanism. We have experimentally documented these effects and included detailed results in the supplementary information.

Comments 6: Please justify the necessity of performing fluorescence spectral characterization since absorption spectra with calibration curves appear sufficient for individual molecular quantification, while the fluorescence data exhibit more complex patterns without demonstrating superior performance in either detection precision or dynamic range compared to absorption measurements. A detailed explanation for including fluorescence characterization should be provided.

Response 6: We thank the reviewer for this insightful comment. It is true that absorption spectra are generally simpler to interpret; however, this point deserves further clarification. Figures 2a and 2b show the absorption spectra of FMN and NADH. It is important to note that NADH is usually present at significantly higher concentrations than FMN. When scanning the absorption spectrum from 500 nm to 300 nm in a clinical sample, the spectrum of NADH tends to dominate and effectively “mask” the absorption features of FMN in the 300–400 nm region, as both molecules share absorption bands in this range. Depending on their relative concentrations, the resulting combined absorption spectrum may assume more or less complex shapes, making it difficult to deconvolute the individual contributions—especially at low FMN concentrations.

In contrast, fluorescence spectroscopy offers higher specificity for FMN detection. When excited at 455 nm, only FMN emits fluorescence between 500–600 nm, as NADH does not absorb at 455 nm. This enables a clean, interference-free signal suitable for FMN quantification. On the other hand, if excited at 340 nm, both NADH and FMN fluoresce: NADH emits across the 400–600 nm range, and FMN contributes with two fluorescence components—one due to its direct excitation at 340 nm, and another due to secondary excitation by light at 455 nm emitted by NADH (cross-excitation). Therefore, while absorption spectroscopy remains suitable for NADH quantification, we demonstrate that fluorescence is more appropriate and selective for FMN detection, especially when combined with appropriate excitation wavelengths.

Additionally, as noted above, the absorption spectra of FMN+NADH mixtures can become highly complex depending on the concentration ratios. However, this complexity can be addressed through careful calibration and the use of supervised machine learning algorithms, which we show can effectively distinguish the individual contributions of FMN and NADH in absorption data.

In light of the reviewer’s observations, we have revised the conclusions to clarify the rationale for including fluorescence characterization in our study and to better articulate the complementary roles of absorption and fluorescence spectroscopy in coenzyme detection.

Comments 7: The manuscript claims in line 460 that previous studies on NADH had a narrow concentration range and missed saturation effects. However, the saturation effect observed in this study occurs at concentrations above 100 μg/mL, making higherconcentration measurements meaningless. Positioning the saturation effect as a breakthrough compared to literature limitations appears inappropriate and requires reconsideration.

Response 7: We thank the reviewer for this important observation. In the cited literature, NADH calibration was typically performed up to concentrations of approximately 30–40 μg mL⁻¹, particularly in studies focused on kidney perfusion. However, in the case of liver perfusion—both hypothermic and normothermic—the physiological context is notably different. The liver is a substantially larger organ with a higher cellular density and metabolic activity, and previous reports have indicated that NADH concentrations can exceed 100 μg mL⁻¹ in clinical samples.

For this reason, we considered it appropriate and necessary to investigate a broader concentration range in our study, including higher values. This allowed us to characterize potential saturation effects in fluorescence detection that would not be observable at lower concentrations. We agree with the reviewer that these effects may not be relevant in all contexts, but we believe they are of practical significance when working with liver perfusion fluids. We have revised the manuscript text accordingly to clarify the rationale for this choice and to better contextualize our findings in relation to the literature.

Comments 8: The manuscript contains numerous careless writing issues that seriously undermine its academic rigor, including but not limited to: (1) part of Figure 1's caption appears in the wrong place, (2) there are two figures labeled as "Figure 1", (3) significant text duplication in lines 155-179, (4) wrong word "fr" in line 135, (5) repeated word "on" in line 167, and (6) an extra period after "water". These oversights demonstrate a lack of attention to detail expected in scholarly writing and raise significant concerns about the reliability of the reported work. A thorough revision with meticulous proofreading is required before further consideration.

Response 8: We thank the reviewer for pointing out these important issues. We acknowledge our oversight and have thoroughly revised the manuscript to address all the concerns raised. Specifically, we have:

  • Corrected the misplaced portion of Figure 1’s caption,
  • Ensured that all figure numbering is consistent and unique,
  • Eliminated the duplicated text between lines 155–179,
  • Fixed the typographical error on line 135 ("fr"),
  • Removed the repeated word "on" in line 167, and
  • Deleted the extra period following "water".

We sincerely apologize for these errors and have carefully proofread the entire manuscript to ensure clarity, consistency, and academic rigor.

Reviewer 3 Report

Comments and Suggestions for Authors

This paper tested the optical characteristics of two biomarkers in machine perfusion in pre-transplantation. However, the results in this work were based on commercial bench spectrometers. The authors did not implement a ready-to-use system to real-time monitor the biomarkers during machine perfusion. Although the authors claimed their work could provide a theoretical foundation for the future development of related optical sensors, the novelty of this work is not enough for publication in Sensors. Secondly, the authors mentioned that current methods lack an efficient real-time monitoring function. I don’t see how this work can contribute to real-time monitoring in machine perfusion and outperform existing methods in this aspect. Last but not least, the writing of this paper needs improvement. For example, the numbering of the figures looks like a mess. The authors should check for these details before submission.

Author Response

Comments 1: This paper tested the optical characteristics of two biomarkers in machine perfusion in pre-transplantation. However, the results in this work were based on commercial bench spectrometers. The authors did not implement a ready-to-use system to real-time monitor the biomarkers during machine perfusion. Although the authors claimed their work could provide a theoretical foundation for the future development of related optical sensors, the novelty of this work is not enough for publication in Sensors. Secondly, the authors mentioned that current methods lack an efficient real-time monitoring function. I don’t see how this work can contribute to real-time monitoring in machine perfusion and outperform existing methods in this aspect. Last but not least, the writing of this paper needs improvement. For example, the numbering of the figures looks like a mess. The authors should check for these details before submission.

Response 1: Dear Reviewer,

We sincerely thank the reviewer for the critical feedback. We greatly value the opportunity to clarify the scientific rationale and potential relevance of our work. Although this study does not present a ready-to-use real-time monitoring device, we are convinced that it lays essential theoretical and practical foundations for the development of such a system, especially for applications in organ perfusion prior to transplantation, and could be useful for the scientific community working in the field.

To illustrate this point clearly, let us consider a practical example: suppose we aim to develop an optical sensor to detect FMN and NADH in the perfusate of an organ preservation system. The first question would be: which optical technique should we use—fluorescence or absorbance? This manuscript provides an evidence-based answer.

Figures 8 and 9 show that fluorescence is a highly effective method for detecting FMN, as it exhibits linear behavior and can be excited at 450–455 nm with minimal interference from NADH. Conversely, fluorescence is less suitable for NADH, as it presents saturation effects at concentrations above ~100 μg/mL, regardless of excitation wavelength. Absorbance, on the other hand, shows a linear response for NADH even at higher concentrations. This leads to a crucial design recommendation: a dual-mode optical system should be implemented, enabling both absorbance and fluorescence detection. Given the geometric differences in optimal optical path design (90° for fluorescence, 180° for absorbance), this has non-trivial implications for sensor layout.

As noted by Müller et al. (2019) and Huwyler et al. (2023), a first-generation real-time detection system has already been developed. However, in their work, fluorescence was calibrated for NADH without exploring the saturation effect at high concentrations, possibly because their calibration range was narrower than ours. Had they tested concentrations above 100–150 μg/mL, especially in liver perfusion (where NADH can exceed 200 μg/mL), signal saturation would likely have emerged as a limiting factor. This is not just a theoretical concern—it underscores the need to rely on absorbance for NADH detection in specific clinical contexts.

A second critical point is the choice of excitation wavelengths. Our results (Figures 7 and 9) clearly show that optimal excitation occurs at 455 nm for FMN and 340 nm for NADH. This is a key consideration when selecting light sources, especially if LED or laser diodes are to be used. Using non-optimal wavelengths—as done in some literature studies (e.g., 405 nm or 485 nm for FMN)—would require longer integration times to achieve the same signal quality. Furthermore, optimizing the excitation wavelength not only improves the selectivity and signal quality, but also directly reduces the required integration time. This is a key parameter for achieving high-frequency, real-time monitoring without overloading the processing capabilities of portable and cost-effective acquisition systems.

In our ongoing hardware development (not mentioned in this manuscript), we have encountered this exact trade-off: extending the integration time to compensate for suboptimal excitation is not feasible with the microcontroller-based architecture we are using. Therefore, our current hardware design is based on the specific theoretical findings reported in this manuscript.

We also highlight another novel and important aspect: spectral shape analysis. While Müller et al. and Huwyler et al. rely on the area under the curve (AUC) for quantification, this approach can be problematic. If other substances in the perfusate absorb at similar wavelengths or emit overlapping fluorescence, the AUC alone cannot distinguish their contributions. This becomes especially relevant if exogenous substances (e.g., antioxidants, flavonoids, other flavins) are introduced during perfusion. For this reason, we propose the future development of machine learning models capable of resolving overlapping spectra and performing classification/regression based on full-spectrum data. While this manuscript does not implement those algorithms, it provides a theoretical justification for why such approaches will be essential in future sensor development.

As we noted in the conclusion, we are currently working on a real-time sensor prototype inspired by the results presented here. Without this preliminary characterization of spectral behavior, excitation/emission interactions, and concentration-dependent effects, it would have been extremely difficult to make informed engineering choices or to avoid the limitations seen in previous designs. We understand the reviewer’s perspective that a final sensor would be of even greater interest. However, we would respectfully point out that a thorough phenomenological understanding must precede technical implementation. Unfortunately, due to industrial secrecy, non-disclosure agreements and IP protection concerns, we are currently unable to disclose hardware details. Still, this paper forms the scientific basis of the device now under development.

We will also revise the manuscript carefully to address the reviewer’s comment regarding the quality of writing and figure numbering. We apologize for any oversight and will correct all formatting issues in the next revision.

Round 2

Reviewer 1 Report

Comments and Suggestions for Authors

Figure 2c still contains information on the ordinate axis that this is a fluorescence spectrum, but the caption says these are absorption spectra. Could you please clarify?

Author Response

Comments 1: Figure 2c still contains information on the ordinate axis that this is a fluorescence spectrum, but the caption says these are absorption spectra. Could you please clarify?

Response 1: We thank the reviewer for pointing out this oversight. The issue was due to a distraction error on our part. We have corrected it in the revised manuscript and confirm that the figure now accurately represents the absorption spectrum as intended.

Reviewer 2 Report

Comments and Suggestions for Authors

Thank you for your detailed responses to the previous comments. Regarding the revised manuscript, please address the following additional questions for further clarification.

  1. On line 47, the authors stated "achieving precise biomarker quantification without invasive sampling," and their response indicates this reflects the method’s "potential for non-destructive and precise biomarker measurement." However, the term "achieving" implies experimental validation, while the current data only demonstrate potential. This wording is misleading and should be revised.
  2. The manuscript fails to clearly convey its novel findings to readers. Based on the presented data, both FMN and NADH coexist in the solution, making their accurate individual quantification crucial. While existing literature (as cited in the manuscript tables) commonly employs fluorescence spectroscopy for FMN detection - a method consistent with the authors' results - the claimed novelty lies in utilizing absorption spectroscopy for NADH quantification. However, the authors explicitly acknowledge the challenge of distinguishing FMN and NADH via absorption spectroscopy due to their spectral overlap. This raises a fundamental question regarding the study's contribution: does this work ultimately demonstrate that no reliable method exists for simultaneous quantification of both FMN and NADH in mixed solutions, despite extensive experimental efforts? If the absorption spectroscopy can’t simultaneously quantify FMN and NADH, what’s the purpose for extensive experiments in different buffers and different conditions?
  3. There appears to be a significant discrepancy between the FMN concentrations reported in line 125 and those utilized in subsequent experimental sections. 

Author Response

Comments 1: On line 47, the authors stated "achieving precise biomarker quantification without invasive sampling," and their response indicates this reflects the method’s "potential for non-destructive and precise biomarker measurement." However, the term "achieving" implies experimental validation, while the current data only demonstrate potential. This wording is misleading and should be revised.

Response 1: We thank the reviewer for this valuable clarification. We agree that the term “achieving” may inadvertently suggest that precise biomarker quantification without invasive sampling has already been experimentally validated in clinical settings. Since our study demonstrates the potential and methodological basis for such quantification, rather than its full clinical implementation, we have revised the sentence accordingly. Specifically, we have replaced “achieving” with “proposing” to more accurately reflect the scope of our work. The corrected sentence now reads:

“[The method’s robustness was validated through comparative studies in clinically relevant solutions, proposing a strategy for precise biomarker quantification without invasive sampling.]”

Comments 2: The manuscript fails to clearly convey its novel findings to readers. Based on the presented data, both FMN and NADH coexist in the solution, making their accurate individual quantification crucial. While existing literature (as cited in the manuscript tables) commonly employs fluorescence spectroscopy for FMN detection - a method consistent with the authors' results - the claimed novelty lies in utilizing absorption spectroscopy for NADH quantification. However, the authors explicitly acknowledge the challenge of distinguishing FMN and NADH via absorption spectroscopy due to their spectral overlap. This raises a fundamental question regarding the study's contribution: does this work ultimately demonstrate that no reliable method exists for simultaneous quantification of both FMN and NADH in mixed solutions, despite extensive experimental efforts? If the absorption spectroscopy can’t simultaneously quantify FMN and NADH, what’s the purpose for extensive experiments in different buffers and different conditions?

Response 2: We thank the Reviewer for this crucial and insightful observation. We recognize that the current presentation may have led to some confusion regarding the novelty and contribution of our work. We clarify below the rationale and significance of our results, responding point-by-point to the Reviewer’s two key questions.

1) Does this study ultimately demonstrate that no reliable method exists for the simultaneous quantification of FMN and NADH in mixed solutions, despite extensive experimental efforts?

Our intention was not to state that there is no method more reliable than another, but rather to highlight the challenges and limitations associated with the simultaneous detection of FMN and NADH using optical spectroscopy, and to propose a practical solution based on their individual spectroscopic properties.

Specifically, we demonstrate that fluorescence is not suitable for NADH quantification due to the clear saturation effect observed at higher concentrations (>100 µg/mL), which is consistent with the physiological range in large organs such as the liver. For this reason, we recommend absorption spectroscopy for NADH quantification, since its calibration curve remains linear and reliable even at high concentrations.

Although FMN and NADH share an overlapping absorption region (300–410 nm), NADH is typically present at much higher concentrations than FMN. Therefore, the absorbance signal is dominated by NADH, as shown in Figures 2a and 2c. The FMN contribution introduces some uncertainty, but does not prevent a useful estimation of NADH concentration. This is why we suggest that clinicians using absorbance methods should not rely solely on raw spectra, but instead apply spectral analysis techniques (e.g., machine learning) to improve interpretation.

In clinical practice, absolute precision is often less critical than detecting general concentration trends. Thus, even if NADH absorbance measurements involve a wider confidence interval, they remain clinically meaningful. Our contribution lies in clarifying these limitations and offering practical guidance for clinical implementation, rather than claiming a definitive solution.

2) If absorption spectroscopy can’t accurately distinguish both, what is the purpose of the extensive experiments in different buffers and conditions?

Our experimental work was designed to systematically study how FMN and NADH behave under different conditions, to identify the sources of uncertainty and optimize measurement strategies. By exploring multiple buffers, pH levels, temperatures, and concentrations, we aimed to understand how these factors affect spectral behavior and how they might be leveraged to enhance selectivity or accuracy.

For instance, in Figure 14 we present representative absorption spectra from a solution containing maximum FMN concentration (0.96 µg/mL) and minimum NADH concentration (20 µg/mL), illustrating how the FMN contribution may influence the accuracy of NADH quantification. Although the spectral overlap limits absolute selectivity, our results indicate that the method can still produce clinically acceptable estimations, particularly when supported by appropriate analytical techniques. In the specific case shown, FMN exerts only a minor influence on the NADH absorbance spectrum; however, depending on the relative concentrations, its impact may become more significant. Ultimately, it is up to the clinician to assess whether the potential margin of error falls within an acceptable diagnostic threshold.

Following the Reviewer’s recommendation, we have revised the conclusions of the manuscript to clearly reflect these points and to better highlight the actual scope and contribution of our work.

Comments 3: There appears to be a significant discrepancy between the FMN concentrations reported in line 125 and those utilized in subsequent experimental sections.

Response 3: We thank the supervisor for pointing out this error of distraction and inconsistency, and we have corrected it.

Reviewer 3 Report

Comments and Suggestions for Authors

In the revised manuscript, the authors better show the importance of their work and make it more convincing. The overall quality of the revised manuscript is improved. However, some mistakes still exist. For example, there are two "Table 1" in the revised version. These "tiny" mistakes make the work look unprofessional. Such typos should be checked and revised before acceptance.

Author Response

Comments 1: In the revised manuscript, the authors better show the importance of their work and make it more convincing. The overall quality of the revised manuscript is improved. However, some mistakes still exist. For example, there are two "Table 1" in the revised version. These "tiny" mistakes make the work look unprofessional. Such typos should be checked and revised before acceptance.

Response 1: We thank the reviewer for the careful reading of our revised manuscript and for the helpful observations. We apologize for the oversight regarding the table numbering and have corrected these distraction errors in the current version. We sincerely appreciate the reviewer’s time and effort in helping us improve the quality and clarity of our work.
